# The Association between the Abundance of Homozygous Deleterious Variants and the Morbidity of Dog Breeds

**DOI:** 10.3390/biology13080574

**Published:** 2024-07-29

**Authors:** Sankar Subramanian, Manoharan Kumar

**Affiliations:** 1Centre for Bioinnovation, University of the Sunshine Coast, Sippy Downs, QLD 4556, Australia; 2School of Science, Technology, and Engineering, University of the Sunshine Coast, Moreton Bay, QLD 4502, Australia; 3Centre for Tropical Bioinformatics and Molecular Biology, James Cook University, Cairns, QLD 4502, Australia; manoharan.kumar@jcu.edu.au

**Keywords:** inbreeding, morbidity, deleterious SNVs, RoH, population bottleneck, small populations

## Abstract

**Simple Summary:**

It is well known that highly inbred dogs are more prone to diseases than less inbred or outbred dogs. This is because inbreeding increases the number of bad mutations present in both paternal and maternal chromosomes (recessive mutations) of the dogs. Using the genome data from 392 dogs belonging to 83 breeds, we investigated the association between the abundance of recessive bad mutations and dog health. The frequency of visits to veterinary clinics for non-routine care was used as the measure of dog health. Our results revealed a highly significant positive relationship between the number of recessive harmful mutations and the degree of dog health. The dog breeds that required more veterinary care had two times higher harmful mutations than those that required less care. The results of this study could be useful for understanding the disease burden on breed dogs and as a guide for dog breeding programs.

**Abstract:**

It is well known that highly inbred dogs are more prone to diseases than less inbred or outbred dogs. This is because inbreeding increases the load of recessive deleterious variants. Using the genomes of 392 dogs belonging to 83 breeds, we investigated the association between the abundance of homozygous deleterious variants and dog health. We used the number of non-routine veterinary care events for each breed to assess the level of morbidity. Our results revealed a highly significant positive relationship between the number of homozygous deleterious variants located within the runs of homozygosity (RoH) tracts of the breeds and the level of morbidity. The dog breeds with low morbidity had a mean of 87 deleterious SNVs within the RoH, but those with very high morbidity had 187 SNVs. A highly significant correlation was also observed for the loss-of-function (LoF) SNVs within RoH tracts. The dog breeds that required more veterinary care had 2.3 times more homozygous LoF SNVs than those that required less veterinary care (112 vs. 50). The results of this study could be useful for understanding the disease burden on breed dogs and as a guide for dog breeding programs.

## 1. Introduction

Dogs were domesticated from wild wolves over 15,000 years ago [1,2,3]. The process of domestication involves only a small number of individuals, which might have resulted in one or more bottlenecks in the founding dog population [4,5,6]. Although dogs might have been domesticated several thousand years ago, modern dog breeds were formed only in the past 300 years through artificial selection and inbreeding [7,8]. Population genetic theories suggest that inbreeding reduces genetic diversity and increases homozygosity and deleterious variants—all of which lead to a reduction in the fitness of dogs [9]. Several empirical studies provide evidence for these theories. Previous studies comparing wild wolves and domesticated dogs showed that the genomic diversity was much reduced for the latter compared to the former and the reduction was much higher for breed dogs than that of village dogs [6,10]. The genomes of breed dogs were found to have a higher proportion of runs of homozygosity (RoH) relative to village dogs and wolves [10]. The accumulation of deleterious variants in domesticated dogs was evidenced by the higher proportion of amino acid-changing variants in breed dogs than in village dogs and wolves [6,10,11,12]. Furthermore, homozygous deleterious and loss-of-function (LoF) variants were also enriched in breed dogs than wolves [6,10]. The fraction of the genome under RoH (FROH) varied significantly between dog breeds, and the FROH of highly inbred dogs was much higher compared to that of low inbred or outbred dogs [10,13,14,15,16,17]. Importantly, deleterious homozygous variants were found within RoH segments, which therefore increased the recessive genetic load of highly inbred dogs [16].

The phenotypic effects of inbreeding have been documented through the observation of deformities and diseases in highly inbred or purebred dogs [18,19,20,21,22,23,24,25,26,27]. For instance, diseases such as cardiomyopathy, epilepsy, hypothyroidism, and dysplasia have been reported to be more prevalent in purebred dogs than in mixed-bred or outbred other dogs [18,19,23,26,27]. Furthermore, several types of cancer have been diagnosed in inbred dogs [27,28,29], and purebred dogs are diagnosed with cancer at a much younger age than mixed-bred dogs [27]. An inverse relationship between body size and life expectancy of breed dogs was reported [30,31]. However, later studies suggested that highly inbred dogs are predominantly large, and hence, the correlation is due to the level of breeding rather than body size [30].

Although earlier studies showed the difference in the number of homozygous deleterious variants among dog breeds, their impact on dog health is unclear. The question is whether dogs with fewer deleterious variants are healthier than those with more such variants. Using pet insurance data, a previous study showed that the level of inbreeding correlates with the level of morbidity [32]. Following this, we used the number of non-routine veterinary care events (VCEs) to measure the level of morbidity (see Section 2) and examined its relationship with the abundance of homozygous deleterious and LoF SNVs present within the RoH segments.

## 2. Materials and Methods

### 2.1. Genome Data

The whole-genome data and body size estimates for 722 canines were obtained from a previous study [33]. After excluding wolves, village dogs, and other canines, the data for 539 breed dogs were available. Out of these, the pet insurance data were available only for 392 dogs belonging to 83 distinct breeds (Appendix A). Most of the genomes had >15X coverage, five of them had 10X–15X, and one had a coverage of 8X (Appendix A). We also included the genome of a coyote, which was used as an outgroup. We excluded indels and retained only the bi-allelic Single Nucleotide Variants (SNVs). The homozygous nucleotides of the coyote genome were used to determine the orientation of mutational changes and identify the derived SNVs.

### 2.2. Morbidity Data

The morbidity values used in this study are the number of non-routine veterinary care events per 10,000 dog years at risk (DYAR), which were obtained from a previous study [32]. The original insurance data belong to Agria Pet Insurance Ltd. (Buckinghamshire, UK; https://www.agriapet.co.uk/, accessed on 23 May 2024) and were accessed through the International Partnership for Dogs (http://dogwellnet.com/, accessed on 23 May 2024). The data belong to the dogs insured during the years 2011–2016. A non-routine veterinary care event indicates that the insurance claim was made for a diagnosis that was neither for preventive care nor for prophylactic measures. Note that multiple VCEs for the same diagnosis were counted only once. The measure DYAR denotes the number of years a dog was insured. For example, if a dog was insured for ten years, the DYAR would be 10. Therefore, morbidity scores for each breed were the VCEs normalised using DYAR. The morbidity values for different dog breeds ranged between 794 to 2439. We grouped the breeds into four categories, low, medium, high, and very high, based on their morbidity scores of <1300, 1300–1600, 1600–1900, and >1900, respectively. The number of breeds in these categories is 11, 31, 26, and 15, respectively.

### 2.3. Analysis

The number of runs of homozygosity segments was estimated using the software Plink verion 1.9 [34] to identify the homozygous segments that are >1 Mb. The software SNPeff verion 3k was used to annotate the genomes and to identify synonymous, nonsynonymous, and intron SNVs [35]. To identify deleterious missense alleles, we used the SIFT score [36], and biallelic variants with a score ≤0.05 were designated as ‘deleterious’. Loss of function (LoF) SNVs were identified based on the annotations “stop_lost”, “stop_gained”, “start_lost”, “splice_donor”, and “splice_acceptor”. Using the information on the genomic boundaries of RoH and the genomic locations of SNVs, we identified the deleterious and LoF SNVs within RoH segments. The number of LoF and deleterious SNVs per genome, along with the standard errors, were estimated for each genome and were averaged for each breed.

The significance between the mean counts was determined using the Z test, and the statistical significance was determined using the software Z to P (http://vassarstats.net/tabs_z.html, accessed on 23 May 2024). A Pearson correlation coefficient was used to determine the strength of the correlation. Furthermore, using the non-parametrical Spearman’s correlation also produced a similar strength of correlation. The first-order partial correlations were calculated through an online web server (http://vassarstats.net/par.html, accessed on 23 May 2024) using the bivariate correlation coefficients obtained for pairwise comparisons. The statistical significance of the correlation was determined by converting the correlation coefficient r to the normal deviate Z, and this was accomplished using the online software r to P (http://vassarstats.net/tabs_r.html, accessed on 23 May 2024). Note that the correlations were performed using the counts of SNVs averaged for single breeds, and the mean estimates of SNV counts were also calculated by grouping the breeds into four morbidity categories.

## 3. Results

To assess the health of the dog breeds, we measured the level of morbidity in terms of DYAR (see Section 2 for details). First, the whole-genome diversity was estimated for each breed and plotted against morbidity, and a highly significant negative relationship (r = −0.42, *p* = 0.0009) was observed between the two variables, suggesting that the breeds with low diversity had a high morbidity score (Table 1). For instance, the heterozygosity of the genomes of breeds with low morbidity was ~0.001, whereas it was 0.0007 for breeds with very high morbidity, and the difference between them was highly significant (*p* = 0.00002, Z test) (Appendix A). Next, we estimated the proportion of genome under runs of homozygosity (FRoH) for breed dogs and averaged them for each dog breed (Appendix A). The regression analysis between FRoH and the level of morbidity showed a highly significant positive correlation (r = 0.45, *p* = 0.0002) (Table 1). While FRoH was 24% of the genomes of the breeds with very high morbidity, it was only 10% of the genomes of the breeds with low morbidity (Figure 1).

We then obtained the counts of deleterious variants present within the RoH fragments of breed dogs and computed the mean for each dog breed (Appendix A). These mean counts were plotted against the morbidity scores, which produced a highly significant positive relationship (r = 0.43, *p* = 0.0005) (Figure 2A and Table 1). The breeds with low morbidity or those requiring less care (fewer VCEs) had a mean of 87 deleterious SNVs, but in contrast, those with very high morbidity or requiring very high care had 187 deleterious SNVs, which is 2.1 times higher than the former (Figure 2B). A similar, highly significant positive correlation was observed for LoF SNVs as well (r = 0.45, *p* = 0.0002) (Figure 2A and Table 1). The very-high-morbidity dog breeds had 2.3 times higher LoF SNVs than the low-morbidity ones (112 vs. 48) (Figure 2B). For these analyses, we used 1 Mb as a threshold to define RoH fragments. However, the strengths of correlations were the same when smaller (0.5 Mb) and larger (2 Mb) thresholds were used (r = 0.47, *p* = 0.0001). In contrast, the magnitude of the difference between the counts of deleterious SNVs of very high and low morbidity breeds was 2.1 times and 2.7 times for 0.5 Mb and 2 Mb RoH fragments, respectively.

Previous studies have shown that the body size and weight of breed dogs correlate with life expectancy, and dogs with large body sizes have low life expectancy [31]. This suggests that the morbidity of large dogs could be higher than that of small dogs. Therefore, we obtained the body weight and height for all dogs and calculated the mean for each breed. Using these data, we performed a partial correlation analysis to control the effects of height and body size. Table 1 shows that correlations between morbidity and other variables such as genome diversity, proportion of RoH fragments, homozygous deleterious, and LoF SNVs were still highly significant (at least, *p* < 0.003) after controlling for the height and body weight of the dog breeds.

## 4. Discussion

Population genetic theories predict that inbreeding reduces the fitness of an organism [37]. The cost of creating purebreds through inbreeding on dog health has been well recognised. For example, the UK Kennel Club has banned mating among first cousins of dogs (https://theconversation.com/how-serious-is-inbreeding-in-show-dogs-56402, accessed on 23 May 2024). The deleterious effects of inbreeding have been inferred by comparing the life expectancy and frequency of diseases among highly inbred, mixed, and outbred dogs [18,19,23,26,27,30,31]. An earlier study provided the first empirical evidence by using pet insurance data and showed that highly inbred dogs have higher morbidity than low inbred or outbred dogs [32]. These data provided the frequency of visits to veterinary clinics, which was available for hundreds of breeds. The current study investigated the correlation between the level of veterinary care required for each breed and the number of homozygous deleterious and LoF SNVs located within RoH segments of these breeds.

Studies in the past showed that genomic signatures such as heterozygosity, runs of homozygous tracts, and homozygous deleterious load vary significantly among dog breeds [6,10,11,12,13,14,15,16,17]. While the first two are only evidence for the level of inbreeding, the third—deleterious load could potentially predict the health of dogs. However, the deleterious and LoF variants were only predicted using statistical and computational methods such as SNPeffect [35], SIFT [36], or other methods but not based on clinical data. The present study addresses this missing link and connects the counts of variants predicted using these methods with the health of dogs through the number of veterinary care events.

In this study, we showed that the dogs experiencing a very high number of VCEs had the largest number of deleterious and LoF variants. There are two possibilities that could explain this. The elevated number of these harmful variants could be associated with many different diseases, and hence, veterinary diagnosis was required for each ailment that could have inflated the number of VCEs. Alternatively, many deleterious variants could be associated with one or a few diseases, which could have increased the severity of the disease, which also might have resulted in a high number of VCEs. The elevated number of deleterious mutations could have resulted from weak selection that was not strong enough to remove them. The weak selection could be due to the small effective size of the founding members of the breeds and also owing to intensive inbreeding.

Clinical studies on dog health have identified genetic variants associated with specific diseases, and many of these studies have also reported homozygous recessive causal variants for those diseases [20,21,22,23,24,25,26]. In contrast, the present investigation showed the potential association between the overall abundance of harmful mutations and dog health. Therefore, the results of this study could be useful in understanding the disease burden on different dog breeds and also useful as a guide for dog breeding programs.

## Figures and Tables

**Figure 1 biology-13-00574-f001:**
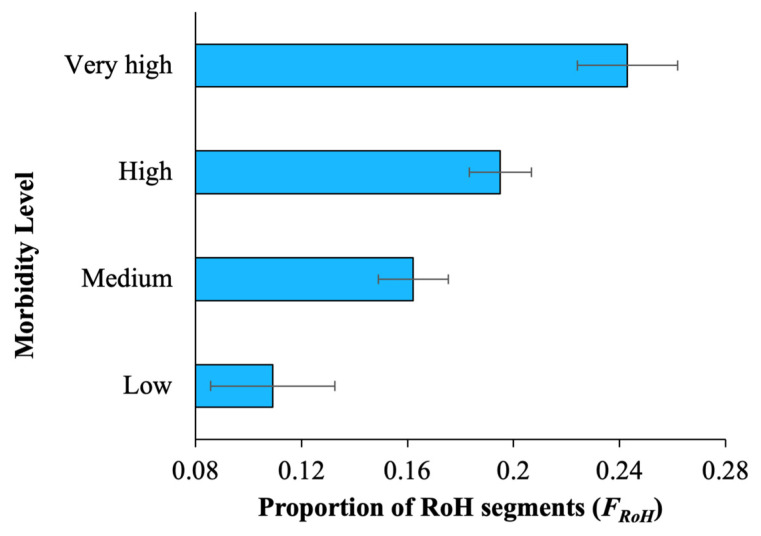
The mean proportions of runs-of-homozygosity segments (FRoH) estimated for the dog breeds belonging to different morbidity levels, which were measured in terms of the number of non-routine veterinary care events (VCEs) per 10,000 dog years at risk (DYAR) (see Section 2). Error bars denote the standard error of the mean.

**Figure 2 biology-13-00574-f002:**
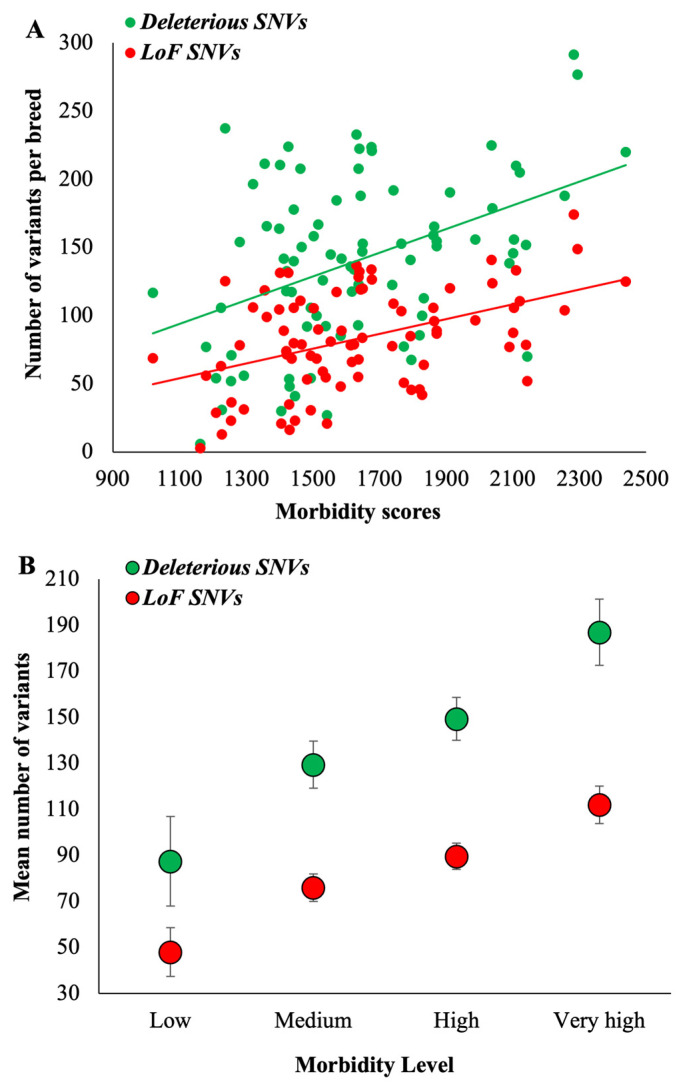
(**A**) Correlation between morbidity scores (number of VCEs/DYAR) and the number of homozygous deleterious or loss-of-function SNVs (located within RoH segments) per dog breed. The relationships are highly significant (*p* < 0.0006) (see Table 1). (**B**) The mean number of homozygous deleterious or loss-of-function SNVs (within RoH) was estimated for the dog breeds belonging to four morbidity groups. The error bars show the standard error of the mean.

**Table 1 biology-13-00574-t001:** Partial correlation analysis to control the effects of height and body weight on morbidity.

Variable 1	Variable 2	Control Variable	*N*	*r*	*p*
Genomic diversity	Morbidity	None	83	−0.42	0.00009
Proportion of RoH segments (*F_RoH_*)	Morbidity	None	83	0.45	0.00002
Deleterious homozygous SNVs	Morbidity	None	83	0.43	0.00005
LoF homozygous SNVs	Morbidity	None	83	0.45	0.00002
Weight	Morbidity	None	78	0.57	<0.000001
Height	Morbidity	None	78	0.35	0.00143
Genomic diversity	Morbidity	Weight	78	−0.39	0.00048
Proportion of RoH segments (*F_RoH_*)	Morbidity	Weight	78	0.36	0.00143
Deleterious homozygous SNVs	Morbidity	Weight	78	0.30	0.00783
LoF homozygous SNVs	Morbidity	Weight	78	0.34	0.00205
Genomic diversity	Morbidity	Height	78	−0.37	0.00086
Proportion of RoH segments (*F_RoH_*)	Morbidity	Height	78	0.38	0.00069
Deleterious homozygous SNVs	Morbidity	Height	78	0.35	0.00174
LoF homozygous SNVs	Morbidity	Height	78	0.38	0.00068

*N* = sample size; *r* = Pearson correlation coefficient.

## Data Availability

This present study did not generate any new data and only used the genome data available from public repositories and previous studies.

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
