# Peer review of "The Association between the Abundance of Homozygous Deleterious Variants and the Morbidity of Dog Breeds"

_biology, 2024, doi:10.3390/biology13080574_

Round 1
Reviewer 1 Report
Comments and Suggestions for Authors
This paper is one of a series from several labs that have looked at how the average morbidity of individuals within a given canine breed or a group of breeds is linked either to inbreeding coefficients determined from pedigree information or to more directly determined genetic factors. Several of these use single nucleotide variation measurements either from genetic testing, GWAS or more recently whole genome sequence data. This paper is interesting in linking non-routine veterinary visit data to whole genome sequence across many breeds to look both at SNP variation and at homozygsity and consequences of the variants seen in a species in which selection in domestication has eliminated many dominant mutations but recessive ones have been conserved in greater numbers.
However there are problems that prevent readers understanding precisely how to interpret all the outcomes of the paper.
line 92. Please briefly justify the choice of >1mb as the cut off for long RoH either in the methods or in the discussion. [You show that in breeds with low morbidity RoH occupy 0.417 times as much of the genome as they do in very high morbidity breeds, but deleterious SNVs in these long RoH are present at a relatively slightly higher level (0.465) in the same comparison, with LoF at an intermediate level between these two figures. Is this difference because there is direct/deliberate selection on alleles that are in fact causative of morbidity. At your discretion you could add some information about whether the proportion of RoH in each genome and of deleterious or LoF SNV as the minimum length of an RoH included in the SNV surveys is reduced. Whether shorter RoH are older could be discussed.]
line 113 ff. In the results section you point out that SNP heterozygosity is about 30% higher in breeds with low morbidity compared to those with very high morbidity. Although there is also a higher proportion of the genome in long runs of homozygosity in the very high morbidity groups it is not clear whether these account for most of this difference in morbidity because of selection at particular areas in the genome, or whether there are also simply more variants from the Dingo and CanFam3 genomes (including more homozygous + heterozygous variants) in breeds with a smaller proportion of long RoH - perhaps likely to be breeds that are older or under lower selection. . In other words is better breed health correlating with higher numbers of variants from a more diverse ancestry, or with less selection (mostly towards particular conformational of behavioural phenotypes) with the consequent RoFs carrying along homozygosity for disadvantageous alleles in terms of morbidity.
Whether or not you choose to make the addition above, please add as supplementary information, the numbers of breeds and of dogs in each breed that you placed into each morbidity group, and the actual total SNV and detrimental SNV numbers for each breed, as well as these SNV within RoH.
In the discussion you should add sentences probably at the bottom of paragraph one to briefly discuss the use of a genetic data set collected from sources including healthy show and pet dogs to try to correlate the consequences of SNVs in RoH to dogs that have visited vets for diagnosis and treatment of disease.
Figure change - line 125-126 (Figure 1). Given the centering of 0.08 under the Y axis the remaining figure labels on X axis are somewhat misleadingly placed (e.g. low morbidity appears to be at about 0.11, and Vh at about 0.25). Put some ticks on the X-axis to indicate where 0.012, 0.016 etc are actually located.
Comments on the Quality of English LanguageThere are a few minor errors of a typographic or grammatical nature, whilst some statements are imprecise and leave the reader unsure of the group being observed. Elsewhere the quality of English language is generally very good. Please make appropriate changes as below:
line 17-19: "The dog 17 breeds with low morbidity had only 87 deleterious SNVs within the RoH, but those with very high morbidity had 187 SNVs," Presumably these are actually rounded mean numbers across all dog breeds in a given category, and within those breeds across all individuals in each breed. Insert "a mean of" after "had" in each clause. If word numbers are a problem, remove the final part of the sentence (which is...etc). Your readership is smart enough not to need it.
line 42-43 is poor English. Remove "than" and replace with something like "when compared with"
line 70-71 it is not entirely clear to the reader whether the "399 dogs of 84 distinct breeds with pet insurance data" is referring to data that was independently available in database from the Plassais et al. study, or whether this figure was the entire dataset of the Bannasch et al. study, or whether (as I suspect) you mean the number of dogs with morbidity data available in the Bannasch et al. study that overlapped with the breeds in the Plassais et al. genetic data. Please rewrite this in a way that leaves no room for doubt.
lines 113- 124: In this paragraph it was again difficult to decide when you were giving information on individual breeds or on groups of breeds. Figure 1 clearly deals with groups of breeds but this need to be stated. In Table 1 I suspect you are dealing with single breeds. Please change or add words in the text to give clarity about this, as you have with the legend to Figure 2.
lines 137-187 deleterious SNV138 "require" not "required" and again add that each breed had "a mean of" 87 / 187 deleterious SNVs
line 171 "showed that". Showed that what? "That" may be surplus to requirements and should be replaced by "the", or you may have forgotten to complete the sentence to say what the genomic signatures showed.
line 201ff: References are double numbered as 1. [1] Davis,...etc. Remove one set of numbers.
Author Response
Reviewer 1
Comment: line 92. Please briefly justify the choice of >1mb as the cut off for long RoH either in the methods or in the discussion. [You show that in breeds with low morbidity RoH occupy 0.417 times as much of the genome as they do in very high morbidity breeds, but deleterious SNVs in these long RoH are present at a relatively slightly higher level (0.465) in the same comparison, with LoF at an intermediate level between these two figures. Is this difference because there is direct/deliberate selection on alleles that are in fact causative of morbidity. At your discretion you could add some information about whether the proportion of RoH in each genome and of deleterious or LoF SNV as the minimum length of an RoH included in the SNV surveys is reduced. Whether shorter RoH are older could be discussed.]
Response: It is a good point. We found that the correlations were strong irrespective of the RoH lengths. But the magnitude of difference (between the counts of deleterious SNVs of breeds with very high and low morbidity) increased with the length of RoH. We have discussed this in lines 145-150.
Comment: line 113 ff. In the results section you point out that SNP heterozygosity is about 30% higher in breeds with low morbidity compared to those with very high morbidity. Although there is also a higher proportion of the genome in long runs of homozygosity in the very high morbidity groups it is not clear whether these account for most of this difference in morbidity because of selection at particular areas in the genome, or whether there are also simply more variants from the Dingo and CanFam3 genomes (including more homozygous + heterozygous variants) in breeds with a smaller proportion of long RoH - perhaps likely to be breeds that are older or under lower selection. . In other words is better breed health correlating with higher numbers of variants from a more diverse ancestry, or with less selection (mostly towards particular conformational of behavioural phenotypes) with the consequent RoFs carrying along homozygosity for disadvantageous alleles in terms of morbidity.
Response: We think the higher number of homozygous deleterious SNVS is due to weak selection that could have resulted from the small number of founders of the breeds and the subsequent inbreeding. This has been discussed in lines 194-197.
Comment: Whether or not you choose to make the addition above, please add as supplementary information, the numbers of breeds and of dogs in each breed that you placed into each morbidity group, and the actual total SNV and detrimental SNV numbers for each breed, as well as these SNV within RoH.
Response: We have now provided a supplementary file containing this information.
Comment: In the discussion you should add sentences probably at the bottom of paragraph one to briefly discuss the use of a genetic data set collected from sources including healthy show and pet dogs to try to correlate the consequences of SNVs in RoH to dogs that have visited vets for diagnosis and treatment of disease.
Response: We agree and discussed this in lines 176-179.
Comment: Figure change - line 125-126 (Figure 1). Given the centering of 0.08 under the Y axis the remaining figure labels on X axis are somewhat misleadingly placed (e.g. low morbidity appears to be at about 0.11, and Vh at about 0.25). Put some ticks on the X-axis to indicate where 0.012, 0.016 etc are actually located.
Response: Figure 1 has been modified to include tick marks.
Comment: line 17-19: "The dog 17 breeds with low morbidity had only 87 deleterious SNVs within the RoH, but those with very high morbidity had 187 SNVs," Presumably these are actually rounded mean numbers across all dog breeds in a given category, and within those breeds across all individuals in each breed. Insert "a mean of" after "had" in each clause. If word numbers are a problem, remove the final part of the sentence (which is...etc). Your readership is smart enough not to need it.
Response: Revised as suggested.
Comment: line 42-43 is poor English. Remove "than" and replace with something like "when compared with"
Response: Done
Comment: line 70-71 it is not entirely clear to the reader whether the "399 dogs of 84 distinct breeds with pet insurance data" is referring to data that was independently available in database from the Plassais et al. study, or whether this figure was the entire dataset of the Bannasch et al. study, or whether (as I suspect) you mean the number of dogs with morbidity data available in the Bannasch et al. study that overlapped with the breeds in the Plassais et al. genetic data. Please rewrite this in a way that leaves no room for doubt.
Response: This has been clarified.
Comment: lines 113- 124: In this paragraph it was again difficult to decide when you were giving information on individual breeds or on groups of breeds. Figure 1 clearly deals with groups of breeds but this need to be stated. In Table 1 I suspect you are dealing with single breeds. Please change or add words in the text to give clarity about this, as you have with the legend to Figure 2.
Response: This has been clarified in the method section now (Lines 112-114).
Comment: lines 137-187 deleterious SNV138 "require" not "required" and again add that each breed had "a mean of" 87 / 187 deleterious SNVs
Response: Done
line 171 "showed that". Showed that what? "That" may be surplus to requirements and should be replaced by "the", or you may have forgotten to complete the sentence to say what the genomic signatures showed.
Response: This has been rewritten.
Comment: line 201ff: References are double numbered as 1. [1] Davis,...etc. Remove one set of numbers.
Response: Duplicate numbers have been removed.
Reviewer 2 Report
Comments and Suggestions for Authors
GENERAL COMMENTS:
Manuscript entitled “The Association Between the Abundance of Homozygous Deleterious Variants and Morbidity of Dog Breeds” by Sankar et al. provides interesting information regarding the relationship between the number of homozygous deleterious variants in purebred dogs and their health morbidity.
In general, several issues need to be addressed to improve the quality of the manuscript before it can be accepted for publication.
SPECIFIC COMMENTS:
L97: The detailed correlation between the genome data and morbidity data should be added in the manuscript.
L88: Please add the method for normalizing the morbidity scores from single individual to the whole breed group. Also, given the different sample sizes for each dog breed, did the morbidity scores were adjusted when calculated?
L98: The author revealed subsets of deleterious SNVs related to the morbidity, weight, height of dogs, what’s the biological function of the corresponding genes? Is there any candidate genes related to certain traits were discovered?
Author Response
Reviewer 2
Comment: L97: The detailed correlation between the genome data and morbidity data should be added in the manuscript.
Response: The details of the correlations between the genome data and morbidity data (including the correlation coefficients and P-values) are given in Table 1, Figure 2 and in the main text (Methods and Results).
Comment: L88: Please add the method for normalizing the morbidity scores from single individual to the whole breed group. Also, given the different sample sizes for each dog breed, did the morbidity scores were adjusted when calculated?
Response: This has been clarified in lines 101-102.
Comment: L98: The author revealed subsets of deleterious SNVs related to the morbidity, weight, height of dogs, what’s the biological function of the corresponding genes? Is there any candidate genes related to certain traits were discovered?
Response: The present study did not focus on the functions of genes associated with those phenotypes, which will be an interesting paper to work on in the future.